# Effects of *Lactobacillus reuteri* and *Streptomyces coelicolor* on Growth Performance of Broiler Chickens

**DOI:** 10.3390/microorganisms9061341

**Published:** 2021-06-21

**Authors:** Sarayu Bhogoju, Collins N. Khwatenge, Thyneice Taylor-Bowden, Gabriel Akerele, Boniface M. Kimathi, Joseph Donkor, Samuel N. Nahashon

**Affiliations:** 1Department of Agricultural and Environmental Sciences, Tennessee State University, Nashville, TN 37209, USA; sarayubhogoju@gmail.com (S.B.); ckhwaten@tnstate.edu (C.N.K.); ttaylor8@tnstate.edu (T.T.-B.); mgakerele@gmail.com (G.A.); muthurikim@gmail.com (B.M.K.); jydonkor@gmail.com (J.D.); 2Department of Biological Sciences, Tennessee State University, Nashville, TN 37209, USA

**Keywords:** broiler, growth performance, probiotics, *Lactobacillus reuteri*, *Streptomyces coelicolor*

## Abstract

There are well documented complications associated with the continuous use of antibiotics in the poultry industry. Over the past few decades, probiotics have emerged as viable alternatives to antibiotics; however, most of these candidate probiotic microorganisms have not been fully evaluated for their effectiveness as potential probiotics for poultry. Recent evaluation of a metagenome of broiler chickens in our laboratory revealed a prevalence of *Lactobacillus reuteri* (*L. reuteri*) and Actinobacteria class of bacteria in their gastrointestinal tract. In this study *Lactobacillus reuteri* and *Streptomyces coelicolor* (*S. coelicolor*) were selected as probiotic bacteria, encapsulated, and added into broiler feed at a concentration of 100 mg/kg of feed. In an 8-week study, 240 one day-old chicks were randomly assigned to four dietary treatments. Three dietary treatments contained two probiotic bacteria in three different proportions (*L. reuteri* and *S. coelicolor* individually at 100 ppm, and mixture of *L. reuteri* and *S. coelicolor* at 50 ppm each). The fourth treatment had no probiotic bacteria and it functioned as the control diet. *L. reuteri* and *S. coelicolor* were added to the feed by using wheat middlings as a carrier at a concentration of 100 ppm (100 mg/kg). Chickens fed diets containing *L. reuteri* and *S. coelicolor* mixture showed 2% improvement in body weight gain, 7% decrease in feed consumption, and 6–7% decrease in feed conversion ratios. This research suggests that *L. reuteri* and *S. coelicolor* have the potential to constitute probiotics in chickens combined or separately, depending on the desired selection of performance index.

## 1. Introduction

Poultry is widely used as a leaner form of meat and an equitable source of protein all over the world. The practice of rearing broiler chickens in limited spaces increases the occurrence of rapid spread of diseases among poultry flocks, creating a continuous challenge in the poultry industry, as it affects productivity leading to economic losses for the producers. Finding a delicate balance between animal health and productivity when it comes to antibiotic use in poultry while minimizing the chances of creating a ‘superbug’ has become problematic, which has led to intensive efforts to investigate alternatives to antibiotic use.

The administration of probiotics has been chosen to aid in the control of infections in poultry. Probiotics are used to enhance the natural immunological capacity and to increase the growth rate in poultry [1]. Currently, lactic acid bacteria (LAB) are being commercially sold as probiotics for the improvement of animal health. However, there is a need to further explore other potential probiotic organisms that will perform well or better than existing probiotics in the poultry industry.

Several LAB are considered as potential probiotics because of their nonpathogenic and beneficial effects on the host. Beneficial lactic acid bacteria include *Lactobacillus acidophilus*, *Lactobacillus plantarum, Lactobacillus fermentum, Lactobacillus reuteri, Lactobacillus salivarius, Bifidobacterium bifidum, Bifidobacterium animalis, Bifidobacterium longum, Enterococcus faecalis, Enterococcus fecium, Streptococcus cremoris, and Streptococcus salivarius*. In this research, *Lactobacillus reuteri* (*L. reuteri*) was selected as a potential probiotic for broilers. *L. reuteri* is a Gram-positive bacterium mostly found in the gut flora of animals and poultry. It is one of the most well documented probiotic species in the *Lactobacillaceae* family and has been widely used as a probiotic in humans and other animals [2,3,4]. Previous research has shown that *L. reuteri* strains can be found in human feces, breast milk, human vagina and the oral cavity, guinea pigs, rats, pigs, broilers, and sourdough bread. *L. reuteri* has the capacity to produce a variety of antimicrobial substances such as lactic acid, hydrogen peroxide [5], reuterin [6,7] and reutericyclin [8]. Reuterin, which is a potent antimicrobial agent produced by *L. reuteri* helps in shaping and modeling the composition and spatial architecture of the gastrointestinal microbiota [9] and is capable of inhibiting a wide spectrum of microorganisms including gram-positive bacteria, gram-negative bacteria, fungi, and protozoa [10]. *L. reuteri* strains can inhibit in vitro growth of many enteric pathogens including *Escherichia coli*, *Salmonella typhimurium*, *Staphylococcus epidermidis*, *Staphylococcus aureus*, *Helicobacter pylori* and rotavirus [2,11,12].

Along with LAB, there are other bacteria that can be utilized as probiotics. The genus *Streptomyces* are Gram-positive bacteria with high G + C (70%) content and soil-living bacteria with characterized branching filamentous morphology. The exceptional biochemistry of *Streptomyces* allows for the production of secondary metabolites which account for almost half of all known antibiotics [13], and for this reason *Streptomyces* has been extensively recognized as a significant microorganism [14,15,16] the secondary metabolites produced including antibiotics [14], antitumor, antiparasitic, immunosuppressive agents, and enzymes [17]. *Streptomyces* is used as a probiotic mainly in aquaculture because of its unique ability to produce several antimicrobial agents. Das et al. [18] and Augustine et al. [19] found that the genus Streptomyces revealed several promising results as probiotic. One of the most studied strains of *Streptomyces* species is *Streptomyces coelicolor* (*S. coelicolor*), which belongs to the bacterial class called Actinobacteria. For a decade, this group of bacteria has been targeted for research due to their diversity and complex life cycles. Most of these metabolic compounds have important applications in human medicine such as antibacterial, antitumor, and antifungal agents. Also, in agriculture these secondary metabolites act as growth promoters, agents for plant protection, antiparasitic agents, and herbicides [20]. In May 2002, the complete genome sequence of *S. coelicolor* was published [21].

Five different secondary metabolites were identified in *S. coelicolor* and four of these have antimicrobial activity: the colored and visually detected actinorhodin (Act), undecylprodigiosin (Red), methylenomycin (Mmy), and calcium dependent antibiotic (CDA). The secondary metabolite actinorhodin (Act) has become a target for further investigation in *S. coelicolor.* This aromatic polyketide has proved to be the most studied example of all other *Streptomyces* antibiotics with its color being dependent on the pH of the environment: it is blue in neutral and alkaline solutions and red in acidic ones, a characteristic that led to the strain being named *S. coelicolor* [22].

The probiotic properties of *S. coelicolor* have not been studied in animals such as chickens, pigs or cattle, even though it is abundantly used in the production of several antibiotics. Only a few studies have been reported using *S. coelicolor* as probiotic in shrimp aquaculture [23] and in fish aquaculture [24]. In this current research, *S. coelicolor* is being evaluated for the first time in poultry for its potential use as a probiotic organism singularly or in combination with *L. reuteri* in broiler chickens.

## 2. Materials and Methods

### 2.1. Ethics Statement

All sample collections and treatments were conducted strictly in accordance with the protocol approved by the Institutional Animal Care and Use Committee (IACUC) of Tennessee State University, USA (Animal Welfare Assurance # A4472-01).

### 2.2. Encapsulation of Microorganisms

Sub-cultured microorganisms were encapsulated before they were added to the experimental diets. A modified encapsulation protocol was adopted [25]. Briefly, inoculated cultures were centrifuged and added to 5% sterile soy protein and 0.2% alginate solution and mixed gently. To form microcapsules and ionic cross linking, sterile 1 M CaCl_2_ solution was added to the protein solution and left to settle for 30 min at room temperature. To remove excess moisture, the solution was centrifuged at 4000 rpm for 15 min at 4 °C. Formed encapsulated pellets were weighed and stored at 4 °C until further use.

### 2.3. Birds and Dietary Treatments

A total of 240-day old chicks were purchased from Aviagen, Inc. (Huntsville, AL, USA). These birds were randomly assigned to four dietary treatments in a completely randomized design (Table 1). The diets comprised a standard broiler diet (SBD) control, SBD + *L. reuteri* (100 ppm), SBD + *S. coelicolor* (100 ppm), SBD + *L. reuteri* (50 ppm) + *S. coelicolor* (50 ppm). The *L. reuteri* and *S. coelicolor* were encapsulated and added to the feed using wheat middlings (1% of diet) as a carrier. The broiler chickens were fed the SBD [26,27] supplemented with the probiotics in phases. The starter diets were fed from hatch to 3 weeks of age (WOA) and they contained 3200 Kcal metabolizable energy (ME)/kg and 23% crude protein (CP). Grower diets were fed from 4–6 WOA and they contained 3200 Kcal ME/kg and 20% CP. At 7–8 WOA the birds were fed the finisher diet comprising 3200 Kcal ME/kg and 20% CP (Table 1) [28]. All the dietary treatments were replicated three times with 20 birds per replicate and feed was provided in mash form. Both feed and drinking water were provided *ad libitum* throughout the study. Water was provided in hanging bell water fountains throughout the experimentation period. Mortality was monitored and recorded as it occurred.

### 2.4. Management of Experimental Birds and Data Collection

At 1 day old, all experimental birds were wing banded, weighed, and assigned to concrete floor pens (240 × 150 cm) covered with pinewood shavings litter to a depth of 10 cm. Each pen served as a treatment replicate and housed 20 birds. The floor house temperature was maintained at 32 °C for the first week and was gradually reduced by 2.8 °C to a steady temperature of 23.9 °C up to 5 WOA. Thereafter there was no supplemental heating provided to the experimental birds and the room temperature was maintained at 21 °C throughout the study. Birds were provided a 23-h light regimen from hatch to 8 WOA and ventilation within the house was maintained by thermostatically controlled exhaust fans. The experimental birds were reared under standard broiler rearing conditions [27] for 8 weeks and their performance was monitored throughout the study.

Experimental birds were weighed weekly until 8 WOA. Body weight gain (BWG), feed consumption (FC) and feed conversion ratios (FCR) were determined weekly.

### 2.5. Processing and Measurement of Carcass Characteristics

At 8 WOA, a total of 48 birds (12 birds per treatment) representing twenty percent of experimental broiler chickens were randomly selected for evaluation of carcass characteristics. Feed and water were withdrawn 12 h prior to slaughter. The birds were then manually caught and crated in plastic coops such that each coop contained eight birds. These birds were immediately transported to the processing facility, located less than 100 m away. The birds were euthanatized by hand using a conventional unilateral neck cut to sever the carotid artery and jugular vein and bled for 180 s. Birds were scalded for 120 s at 63 °C and picked for 30 s in a commercial in-line picker (Cantrell Model CPF-60, Cantrell Machine Co., Inc., Gainesville, GA, USA). After the head, shanks, feet, and feathers were removed, the carcass was eviscerated manually by cutting around the vent to remove all the viscera including the kidneys. The whole carcass, thighs, drumsticks, wings, breast, and fat were excised from each subsampled experimental bird and weighed. The weights of individual parts were expressed as percent of live body weight.

### 2.6. Statistical Analysis

Data were analyzed by the ANOVA option of GLM of the SAS/STAT software 9.3 [29] with treatments as a main effect. The following statistical model was used for body weight gain, feed consumption, and feed conversion ratio:Y_ijklm_ = μ + T_i_ + T_k_ + (TT) _ik +_ R _ijkl_ + y_ijklm_
where Y_ijklm_ represents the response variables from each individual replication, μ represents the overall mean, T_i_ represents the effect of dietary treatment; T_k_ represents the effect of time in weeks; (TT) _ik_ represents the effect due to interactions between treatment and time; R_ijkl_ represents the inter-experimental unit (replications) error term and y_ijklm_ represents the intra-experimental unit error term. When there is a significant F-value, means were separated using the least squares means option. Differences in mortality among dietary treatments were analyzed using the chi-square method. Significance implies *p* < 0.05, unless stated otherwise.

## 3. Results

Mean feed consumption (FC) of broiler chickens from hatch to 8 WOA is presented in Table 2. As expected, FC increased with the age of the birds. At 1 WOA, broiler chickens fed diets containing only *L. reuteri* had the highest FC compared to all other dietary treatments. Even though feeding the probiotics did not yield a consistent pattern in feed consumption after the first WOA, it was noted that at 2–3 WOA birds fed diets containing either the *L. reuteri* or *S. coelicolor* or a combination of the two probiotics had lower FC (*p* < 0.05) than birds fed the control diet. Overall, cumulative feed consumption from hatch to 4 WOA of birds fed the 100 ppm each of *L. reuteri* and *S. coelicolor* was significantly (*p* < 0.05) lower by 1 and 3%, respectively, when compared to the control. However, differences in the cumulative feed consumption from hatch to 4 WOA of the control broilers was not different from that of the broilers fed diets containing the combination of *L. reuteri* and *S. coelicolor* (1:1, 50 ppm). On the other hand, differences in cumulative feed consumption of birds fed the control and those fed the combination of *L. reuteri* and *S. coelicolor* were not significant at 5–8 WOA (*p* > 0.05). However, cumulative feed consumption from 5–8 WOA of birds fed the 100 ppm of either *L. reuteri* or *S. coelicolor* was significantly (*p* < 0.05) lower by 14 and 7%, respectively, when compared to the control. Overall, the hatch to 8 WOA cumulative feed consumption of the birds fed the 100 ppm *L. reuteri* or *S. coelicolor* was 10 and 6% lower (*p* < 0.05), respectively, than that of birds fed the control diet. Differences in total feed consumption of birds fed the control diets at hatch to 8 WOA and those fed the combination (1:1, 50 ppm) of *L. reuteri* and *S. coelicolor* were not significant (*p* > 0.05).

Body weight gain (BWG) data of broiler chickens fed the probiotics from hatch to 8 WOA is presented in Table 3. Broiler chickens fed *L*. *reuteri* had higher BWG compared to all dietary treatments at 1 WOA, and birds fed *S. coelicolor* at 2 WOA had higher BWG compared to all dietary treatments. At 3–4 WOA birds feed *L*. *reuteri* showed higher BWG compared to all dietary treatments. The hatch to 4 WOA cumulative BWG of broiler chickens fed the combination of *L. reuteri* or *S. coelicolor* (1:1, 50 ppm) and those fed the 100 ppm *L. reuteri* were 2% and 1% higher than the control (*p* < 0.05), respectively. Birds fed with *L*. *reuteri* and *S. coelicolor* combined diet at 5 WOA had the higher BWG compared to birds feed control diet. 

Cumulative BWG at 5–8 WOA of the broiler chickens fed the diets containing either *L. reuteri* or *S. coelicolor* were significantly lower (*p* < 0.05) that those of birds fed the control diet (11–12%) and those fed the combination of *L. reuteri* and *S. coelicolor* (1:1, 50 ppm) by about 8–9%. However, the reduction if BWG of birds fed the combination of *L. reuteri* and *S. coelicolor* was only 2% when compared to the control. Therefore, *L. reuteri* and the combination of *L. reuteri* and *S. coelicolor* improved BWG of broilers at hatch to 4 WOA, but not at 5–8 WOA. The synergistic effect of the *L. reuteri* and *S. coelicolor* seem to ameliorate the depression of BWG observed in feeding 100 ppm of either *L. reuteri* or *S. coelicolor.*

The mean feed conversion ratio (FCR) of broiler chickens fed probiotics from hatch to 8 WOA are shown in Table 4. At 1 WOA, birds fed diets containing either *L. reuteri* or *S. coelicolor* exhibited a higher FCR than birds fed the control diet and those fed the combination of *L. reuteri* and *S. coelicolor* (*p* < 0.05). Differences in FCR between birds fed the control diets and those fed the combined *L. reuteri* and *S. coelicolor* were not significant (*p* > 0.05). Average FCR of broilers fed diets containing the combined *L. reuteri* and *S. Coelicolor* during week 1–4 was 22% lower (*p* < 0.05) than that of broilers fed the control diet. The average FCR of birds fed either the *L. reuteri* or *S. coelicolor* was not different from the control. The average FCR of week 5–8 was 11, 12, and 8% lower in broilers fed diets containing the *L. reuteri, S. coelicolor, and combination of L. reuteri* and *S. coelicolor,* respectively, when compared with the control. On the other hand, the FCR of birds fed either the *L. reuteri* or *S. coelicolor* was 4% lower than that of birds fed the *L. reuteri* and *S. coelicolor* combined.

Carcass yields of broiler chickens fed probiotics from hatch to 8 WOA are presented in Table 5. In this study carcass weights, breasts, drumsticks, wings, and fat were evaluated. The mean differences in percent weight of carcass, breast, thighs, wings and fat among the dietary treatments were not significant (*p* > 0.05). However, the percent breast weight of birds fed diets containing *S. coelicolor* was 4% higher than that of the control and 9% higher than that of *L. reuteri,* and combined *S. coelicolor* and *L. reuteri.* Birds fed either *S. coelicolor* or *L. reuteri* diets exhibited 4 and 1% higher carcass weights than the control, respectively. Even though not statistically significant (*p* > 0.05), abdominal fat percentage was 3% lower in birds fed the diets containing *L. reuteri* when compared to other dietary treatments. Therefore, highest carcass yield was recorded in birds fed diets containing *S. coelicolor* whereas the lowest fat percentage was recorded in broiler chickens fed diets containing *L. reuteri.*

## 4. Discussion

Probiotics have increasingly emerged as viable alternative to antibiotics in poultry production. We conducted a feeding trial to investigate the effects of dietary supplementations of *L. reuteri* and *S. coelicolor* as potential probiotic bacteria on broiler performance. We evaluated feed consumption (FC), body weight gains (BWG), feed conversion ratios (FCRs) and carcass yield percentage to determine the effect of dietary inclusion of *L. reuteri* and *S. coelicolor* on growth performance of broiler chickens. This is the first study in poultry to the best of our knowledge where *S. coelicolor* is being used as a potential probiotic organism in broiler chickens.

Our findings show that dietary inclusion of both *L. reuteri* and *S. coelicolor* had significant effect on performance of broiler chickens. Supplementations of *L. reuteri* and *S. coelicolor* to broiler chicken’s diet at 1 WOA greatly increased feed consumption compared to chickens fed the control diet. By 4 WOA broiler chickens have reached the grower period, in which the protein and calcium requirements for growth decreases and the diet must maintain fast growing birds, yet in the current research the addition of *L. reuteri* and *S. coelicolor* to broiler chicken’s diets slightly increased feed consumption compared to broiler chickens fed the control diet. The results from our study discovered that at 8 WOA birds fed the combination diet of *L. reuteri* and *S. coelicolor* had the highest feed consumption of all other dietary treatments, which correlates with studies by Shim et al. [30] reporting increased weight gain and feed intake in the finisher phase of broiler chicken was due to administration of multi-microbe probiotics formulations. In contrast to our findings, Hung et al. [31] stated that dietary use of the probiotic *B. coagulans* reduced the average daily feed intake by 8% in the broiler grower-finisher phase (days 22–42). Mookiah et al. [32] also demonstrated a decline in feed intake of 5.6% during the starter phase (1–21 days) in birds treated with a multi-strain probiotic containing 11 *Lactobacillus* strains (*L*. *reuteri* C1, C10 and C16; *L. gallinarum* I16 and I26; *L. brevis* I12, I23, I25, I218 and I211, and *L. salivarius* I24). The contrast between our research and previous studies could be due to species and strain of bacteria selected as the targeted probiotics as well as the chosen host animals.

The BWG results from this study are in agreement with the previous studies from Lan et al. [33] Aluwong et al. [34] Shahir et al. [35], which demonstrated in chickens, both probiotics and prebiotics help to increase weight gain and reduce feed conversion ratio. Enhancement in growth performance and feed efficiency of broiler chickens which fed probiotics [36,37,38,39,40,41,42,43,44] is a culmination of probiotic action like maintaining of beneficial microbial population [45], improving feed intake, digestion, [46,47,48] and altering bacterial metabolism [49,50,51]. Previous research also indicate that longer villi were observed in the ileum of chicks and turkeys treated with *Lactobacillus reuteri* [52] and the concentrations of amylase in broiler intestine were improved after supplementation of diet with *Lactobacillus acidophilus* or a mixture of *Lactobacillus* strains [44,53,54]. Body weight results of chickens seem to be in agreement with Wang et al. [55], in weaned pigs with administration of *L. fermentum* I5007 significantly increased weight gain and feed intake compared to control pigs. Our findings are also in agreement with Yu et al. [56], who reported that *Lactobacillus reuteri* produced a modest improvement in weight gain in broiler chickens during 0–6 WOA and Awad et al. [57] who reported that probiotic supplementation significantly improved broiler chicks at the finisher stage. In a similar study, Nakphaichit et al. [58], Bansal et al. [59] and Olnood et al. [60] reported consistent findings; they observed an increase in BW gain in the first WOA when broiler chicks were fed diets containing 5 log cfu/g *Lactobacillus reuteri* KUB-AC5. Our findings from broiler chickens at 1 WOA fed *L. reuteri,* also correlates with the report of Wang et al. [61] which demonstrated that weaned piglets supplemented with *L. fermentum* had increased weight gains and feed intakes compared to piglets fed a control diet during week 1. The correlations from this study may reveal a similar probiotic/host interaction that lactic acid bacteria like *L. reuteri* share, regardless of the species as it pertains to probiotic effects on BWG in broiler chickens and other agricultural animals. In addition, *S. coelicolor* has exhibited an ability to increase body weight gain when incorporated into broiler chicken’s diet which to date has not been documented.

FCR results agree with studies conducted by Wang et al. [55], Yu et al. [56], and Murugesan and Persia [47] as both studies demonstrate enhancement of feed conversion ratios due to the effect of probiotics. Revealing at different time periods, *L. reuteri* and *S. coelicolor,* whether fed separately or combined, had a beneficial effect on broiler chicken FCR.

According to previous research in the U.S. many producers are focused on growth of breast muscle due to its exceptional economic value, when compared to other carcass characteristics [62,63]. Usually, breast muscle represents about 30% of total edible meat of carcass and about 60% of the protein from the carcass [64]. In this study, all the dietary treatments in chickens are not significantly different with each other in their breast yield. Although all the dietary treatments have shown more than 30% breast muscle yield to its carcass, especially in birds fed with *S. coelicolor* recorded the highest breast muscle yield when compared to other treatments. The carcass yield data is in correlation with the studies of Awad et al. [57] and Humam et al. [65] which attributed enhancement of bird performance to probiotics. Although, abdominal fat yield is not statistically different in all the dietary treatments, birds fed with *L. reuteri* have shown the lowest fat percentage when compared to other treatments, this result may be credited to the decreasing effect of probiotics on fat deposition [37,66]. These results are in contrast with the studies of Moreira et al. [67] and Vargas Jr. et al. [68] that found no differences in the carcass yield by administering probiotics, while Rehman et al. [69] concluded that supplementation of prebiotics or probiotics can improve the growth performance of broilers, Yousefi and Karkoodi, [70] and Sarangi et al. [71] reported contradicting findings; weight gain was not affected by supplementation of probiotics in broiler diet. In this study, we discovered that *S. coelicolor* possessed beneficial effects on carcass percentage and breast percentage compared to all the other dietary treatments, while *L. reuteri* possessed positive effects on abdominal fat yield. These findings may have pinpointed specific beneficial functions of *S. coelicolor* providing desirable carcass characteristics to the poultry industry and *L. reuteri* as a probiotic in broiler chickens showcased a role in fat deposition that probiotics may play.

## 5. Conclusions

Dietary inclusion of *L. reuteri* and *S. coelicolor* as probiotic bacteria in broiler diets revealed their effects in the enhancement of broiler chicken performance. Most changes in broiler performance were observed at 0–4 WOA. Even though not statistically significant, feeding broilers diets containing *L. reuteri* and *S. coelicolor* separately or combined decreased their FCR by about 5% and slightly improved their body weight gain. Abdominal fat was decreased by 21% in birds fed *L. reuteri* whereas feeding *S. coelicolor* increased breast weight of the broilers by 9% at 0–4 WOA. From a producers’ economic standpoint these improvements in bird performance can be quite significant. There seems to be a pronounced correlation with our finding and previous studies examining *L. reuteri* in chickens and pigs. We have shown that both *L. reuteri* and *S. coelicolor* have beneficial effects on broiler chickens as it pertains to FC, BWG, FCR, carcass, and breast yield yet in-depth studies need to be performed to access gut health of broiler chickens and their ability to protect from pathogenic bacteria or inhibit their growth. This is the first study to demonstrate the effect of *S. coelicolor* in broiler chickens as a probiotic. To realize the full potential for *S. coelicolor,* further investigation is needed to evaluate further the bacteria as a potential probiotic in other domestic animals.

## Figures and Tables

**Table 1 microorganisms-09-01341-t001:** Composition of experimental diets fed to Chickens from hatch-8 WOA (% diet).

Ingredients	0–3 WOA	4–6 WOA	7–8 WOA
Corn (8% CP)	46.468	56.088	62.000
Soybean meal (47.5%)	40.000	32.000	27.000
Wheat middlings	1.000	1.000	1.000
Alfalfa meal (17% CP)	1.000	1.000	1.000
Poult. Blend. Fat (8158 Kcal ME/Kg)	7.900	6.300	5.388
Dical. Phosphate (18% P, 22% Ca)	1.900	1.900	1.900
Limestone flour (38% Ca)	1.000	1.000	1.000
D,L-Methionine (98%) ^1^	0.150	0.130	0.130
L-Cystenine (98%)	0.032	0.032	0.032
Salt	0.300	0.300	0.300
Vitamin-Mineral premix ^2^	0.250	0.250	0.250
Calculated levels			
ME (Kcal/Kg)	3200	3200	3200
CP (%)	23	20	20
Calcium	0.93	0.91	0.89
Total Phosphorous	0.73	0.70	0.69
Avail Phosphorous	0.47	0.46	0.45
Cysteine	0.40	0.36	0.34
Methionine	0.50	0.44	0.42
Meth+Cys	0.91	0.81	0.76
Arg	1.58	1.34	1.19
Thr	0.89	0.77	0.69
Lys	1.31	1.10	0.97
Analyzed Levels (%)			
Crude Protein	22.92	20.03	20.06
Crude Fat	4.91	5.20	5.51
Calcium	0.90	0.89	0.91

^1^ Degussa Corporation (Kennesaw, GA, USA). ^2^ Provided per kilogram of diet: retinyl acetate, 3500 IU; cholecalciferol, 1000 ICU; DL-α-tocopheryl acetate, 4.5 IU; menadione sodium bisulfite complex, 2.8 mg; vitamin B12, 5.0 mg; riboflavin, 2.5 mg; pantothenic acid, 4.0 mg; niacin, 15.0 mg; choline, 172 mg; folic acid, 230 mg; ethoxyquin, 56.7 mg; manganese, 65 mg; iodine, 1 mg; iron, 54. mg; copper, 6 mg; zinc, 55 mg; selenium, 0.3 mg.

**Table 2 microorganisms-09-01341-t002:** Feed consumption of broiler chickens fed diets containing probiotics from hatch to 8 WOA.

Treatment		(Weeks of Age)								
Lacto ^1^	Strepto ^2^	1	2	3	4	5	6	7	8	Total
(ppm)		---------------------------------------------------------------(g/Bird/Week)---------------------------------------------------------(g)								
0	0	122.30 ^c^	340.75 ^a^	662.21 ^a^	852.87 ^a^	1104.19 ^a^	1120.06 ^a^	1411.04 ^a^	1058.98 ^b^	6672 ^a^
100	0	160.82 ^a^	332.25 ^b^	641.53 ^b^	816.37 ^c^	1012.83 ^d^	1035.94 ^c^	1200.31 ^d^	862.74 ^c^	6063 ^c^
0	100	133.43 ^b^	326.68 ^c^	625.20 ^c^	830.20 ^b^	1026.63 ^c^	1041.24 ^c^	1272.05 ^c^	1064.69 ^b^	6320 ^b^
50	50	139.16 ^b^	324.55 ^c^	641.01 ^b^	860.79 ^a^	1084.62 ^b^	1099.63 ^b^	1350.75 ^b^	1144.31 ^a^	6645 ^a^
PSEM ^3^		2.31	1.75	2.88	2.94	6.08	5.78	4.13	12.19	33.27
Probability		*p* < 0.001	*p* < 0.001	*p* < 0.001	*p* < 0.001	*p* < 0.001	*p* < 0.001	*p* < 0.001	*p* < 0.001	*p* < 0.001

^1^*Lactobacillus reuteri*. ^2^*Streptomyces coelicolor*. ^3^ Pooled standard error of mean. ^a,b,c^ Means within columns with no common superscript differ significantly (*p* < 0.05).

**Table 3 microorganisms-09-01341-t003:** Body weight gains of broiler chickens fed diets containing probiotics from hatch to 8 WOA.

Treatment		Weeks of Age								
Lacto ^1^	Strepto ^2^	1	2	3	4	5	6	7	8	Total
(ppm)		---------------------------------------------------------------(g/Bird/Week)--------------------------------------------------(g)								
0	0	86.73 ^b^	248.28 ^a^	455.38 ^b^	554.44 ^a^	592.52 ^a,b^	579.90 ^a^	732.07 ^a^	344.54 ^a^	3594
100	0	98.25 ^a^	240.65 ^a^	488.06 ^a^	528.87 ^a,b^	570.24 ^b^	545.82 ^a^	600.72 ^b^	290.93 ^a^	3364
0	100	84.49 ^b^	253.65 ^a^	444.13 ^b^	540.85 ^a^	611.22 ^a,b^	483.88 ^b^	573.47 ^b^	360.69 ^a^	3352
50	50	93.88 ^a^	253.56 ^a^	463.57 ^a,b^	558.91 ^a^	625.55 ^a^	531.21 ^a,b^	669.06 ^a,b^	365.65 ^a^	3561
PSEM ^3^		2.37	6.18	10.03	11.11	16.02	19.32	28.66	31.6	-
Probability		*p* < 0.001	*p* < 0.001	*p* < 0.001	*p* < 0.001	*p* < 0.001	*p* < 0.001	*p* < 0.001	*p* < 0.001	-

^1^*Lactobacillus reuteri.*^2^*Streptomyces coelicolor.*^3^ Pooled standard error of mean. ^a,b^ Means within columns with no common superscript differ significantly (*p* < 0.05).

**Table 4 microorganisms-09-01341-t004:** Feed Conversion ratios of chickens fed different probiotics from hatch to 8 weeks of age.

Treatment		(Weeks of Age)								
Lacto ^1^	Strepto ^2^	1	2	3	4	5	6	7	8	Average
(ppm)		(g Feed/g Body Weight Gain)								
0	0	1.43 ^b^	1.42 ^a^	1.52 ^a^	1.56 ^a^	2.05 ^a^	2.04 ^a^	2.45 ^a^	3.43 ^a^	1.99
100	0	1.68 ^a^	1.45 ^a^	1.35 ^b^	1.60 ^a^	1.84 ^a,b^	1.99 ^a^	2.24 ^a^	2.92 ^a^	1.88
0	100	1.66 ^a^	1.34 ^a^	1.45 ^a,b^	1.56 ^a^	1.71 ^b^	2.09 ^a^	2.20 ^a^	2.92 ^a^	1.86
50	50	1.52 ^b^	1.32 ^a^	1.45 ^a,b^	1.58 ^a^	1.78 ^b^	2.16 ^a^	2.18 ^a^	3.12 ^a^	1.88
PSEM ^3^		0.04	0.04	0.05	0.03	0.08	0.07	0.21	0.19	-
Probability		*p* < 0.001	*p* < 0.001	*p* < 0.001	*p* < 0.001	*p* < 0.001	*p* < 0.001	*p* < 0.001	*p* < 0.001	-

^1^*Lactobacillus reuteri.* ^2^*Streptomyces coelicolor.* ^3^ Pooled standard error of mean. ^a,b^ Means within columns with no common superscript differ significantly (*p* < 0.05).

**Table 5 microorganisms-09-01341-t005:** Carcass characteristics of chickens fed different probiotics from hatch to 8 WOA.

Dietary Treatment (ppm)		Carcass Characteristic (%)						
Lactobacillus	Streptococcus	Carcass	Breast	Thighs	Drumsticks	Wings	Fat	Prob ^1^
0	0	81 ^a^	33.9 ^a^	11.8 ^a^	10.1 ^a,b^	8.6 ^a^	1.4 ^a^	*p* < 0.001
100	0	82 ^a^	32.5 ^a^	12.4 ^a^	10.5 ^a^	9.2 ^a^	1.1 ^a^	*p* < 0.001
0	100	84 ^a^	35.3 ^a^	12.0 ^a^	9.5 ^b^	8.7 ^a^	1.5 ^a^	*p* < 0.001
50	50	80 ^a^	32.7 ^a^	11.6 ^a^	10.2 ^a,b^	8.7 ^a^	1.4 ^a^	*p* < 0.001
PSEM ^2^		1.40	0.99	0.43	0.29	0.28	0.17	-

^1^ Probability. ^2^ Pooled standard error of mean. ^a,b^ Means within columns with no common superscript differ significantly (*p* < 0.05).

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
