# Peer review of "Effects of Lactobacillus reuteri and Streptomyces coelicolor on Growth Performance of Broiler Chickens"

_microorganisms, 2021, doi:10.3390/microorganisms9061341_

Round 1

Reviewer 1 Report

In the present paper I reviewed, the Authors have investigated an interesting topic related to the effects of Lactobacillus reuteri and Streptomyces coelicolor on growth performance of broiler chickens. As the Authors stated, this is reported  for the first time in poultry, S. coelicolor is being evaluated for its potential use as probiotic organism singularly or in combination with L. reuteri in broiler chickens.

I would like to congratulate Authors for the good-quality of their article, the literature reported used to write the paper, and for the clear and appropriate structure.

The manuscript is well written, presented and discussed, and understandable to a specialist readership.

In general, the organization and the structure of the article are satisfactory and in agreement with the journal instructions for authors. The subject is adequate with the overall journal scope. The work shows a conscientious study in which a very exhaustive discussion of the literature available has been carried out. The introduction provides sufficient background, and the other sections include results clearly presented and analyzed exhaustively.

However, as specific comment, with the aim to further improve the quality of the paper, the Conclusion section could be improved; also, the Authors have to check if alle references have been cited in the text.

Also, the Table 1 should be revised: for example, Corn Gluten meal (60% CP) may be deleted because of it is not included in diets; the sum of ingredients is >100; avoid the abbreviated words in Table.

So, I recommend the acceptance of the paper after minor revision.

Author Response

Comment 1: I would like to congratulate Authors for the good-quality of their article, the literature reported used to write the paper, and for the clear and appropriate structure.

The manuscript is well written, presented and discussed, and understandable to a specialist readership.

In general, the organization and the structure of the article are satisfactory and in agreement with the journal instructions for authors. The subject is adequate with the overall journal scope. The work shows a conscientious study in which a very exhaustive discussion of the literature available has been carried out. The introduction provides sufficient background, and the other sections include results clearly presented and analyzed exhaustively.

Response: Thank you for your encouraging comments

However, as specific comment, with the aim to further improve the quality of the paper, the Conclusion section could be improved; also, the Authors have to check if all the references have been cited in the text.

Response: The conclusion was revised, making several changes (lines 334-341 of the revised manuscript). The references were also checked and one reference added (25, L422-423 of the revised manuscript).

Also, the Table 1 should be revised: for example, Corn Gluten meal (60% CP) may be deleted because of it is not included in diets; the sum of ingredients is >100; avoid the abbreviated words in Table.

Response: The Table 1 was revised accordingly and ingredients not utilized were removed from the Table. The sum of ingredients is 100%.

Reviewer 2 Report

This study was the first time that S. coelicolor was fed to broilers, and the findings show its potential as a feed additive and can be evaluated as having important information.

However, in order to be published as a paper, the following revisions are required.

L.105 How did you encapsulate the microorganism?

Table1 Match the significant digits with the other tables.
    Why don't you delete the entries with 0?

Table2-5 Remove useless lines.

In conclusion 
There are some expressions that are not statistically appropriate and should be corrected.
In particular, L332-334.

Author Response

This study was the first time that S. coelicolor was fed to broilers, and the findings show its potential as a feed additive and can be evaluated as having important information.

Response 1: Thank you for your critical inputs.

L.105 How did you encapsulate the microorganism?

Response: The encapsulation protocol was added (L103-109 of the revised manuscript). A reference was also added as ref#25 (L422-423 of the revised manuscript).

Table1 Match the significant digits with the other tables.Why don't you delete the entries with 0?

Response: Table 1 was revised accordingly and the unused ingredients deleted.

Table2-5 Remove useless lines.

Response: Most unnecessary lines in Tables 2-5 were removed.

In conclusion 
There are some expressions that are not statistically appropriate and should be corrected.
In particular, L332-334.

Response: The conclusion was revised, making several changes (lines 334-341 of the revised manuscript).

Reviewer 3 Report

Dear all,

Thank you for your interesting work on probiotics. This is a very well-written article. Maybe you could add in the introduction section a general state for " what is considered as probiotic". 

Minors corrections apply in line 44: "as well as", keep only the word "well"

Line 47 and 48 write the first name of each bacteria in full

Line 217 replace "lover" with "lower"

Author Response

Thank you for your interesting work on probiotics. This is a very well-written article. Maybe you could add in the introduction section a general state for " what is considered as probiotic". 

Response: Thank you for your review and comments

Minors corrections apply in line 44: "as well as", keep only the word "well"

Response: The change was made as suggested for L44.

Line 47 and 48 write the first name of each bacteria in full

Response: The first names of each bacteria is now written in full, L47-48.

Line 217 replace "lover" with "lower"

Response: The change was made accordingly.